# Rural-urban differences in health outcomes, healthcare use, and expenditures among older adults under universal health insurance in China

**Meiling Ying**[1]*, **Sijiu Wang**[1], **Chen Bai**[2], **Yue Li**[1]

**1** Department of Public Health Sciences, School of Medicine and Dentistry, University of Rochester, Rochester, NY, United States of America, **2** Department of Social Security, School of Labor and Human Resources, Renmin University of China, Beijing, China

* Meiling_Ying@urmc.rochester.edu

## Abstract

Rural-urban inequalities in health status and access to care are a significant issue in China, especially among older adults. However, the rural-urban differences in health outcomes, healthcare use, and expenditures among insured elders following China's comprehensive healthcare reforms in 2009 remain unclear. Using the Chinese Longitudinal Healthy Longevity Surveys data containing a sample of 2,624 urban and 6,297 rural residents aged 65 and older, we performed multivariable regression analyses to determine rural-urban differences in physical and psychological functions, self-reported access to care, and healthcare expenditures, after adjusting for individual socio-demographic characteristics and health conditions. Nonparametric tests were used to evaluate the changes in rural-urban differences between 2011 and 2014. Compared to rural residents, urban residents were more dependent on activities of daily living (ADLs) and instrumental ADLs. Urban residents reported better adequate access to care, higher adjusted total expenditures for inpatient, outpatient, and total care, and higher adjusted out-of-pocket spending for outpatient and total care. However, rural residents had higher adjusted self-payment ratios for total care. Rural-urban differences in health outcomes, adequate access to care, and self-payment ratio significantly narrowed, but rural-urban differences in healthcare expenditures significantly increased from 2011 to 2014. Our findings revealed that although health and healthcare access improved for both rural and urban older adults in China between 2011 and 2014, rural-urban differences showed mixed trends. These findings provide empirical support for China's implementation of integrated rural and urban public health insurance systems, and further suggest that inequalities in healthcare resource distribution and economic development between rural and urban areas should be addressed to further reduce the rural-urban differences.

**Data Availability Statement:** There are legal restrictions on sharing the data, which is owned by Peking University. However, other researchers may apply for access to the data through The Center for

Healthy Aging and Development Studies of
National School of Development at Peking
University via the following URL: https://opendata.
pku.edu.cn/dataverse/CHADS.

**Funding:** The author(s) received no specific
funding for this work

**Competing interests:** The authors have declared
that no competing interests exist.

## Introduction

Inequitable access to health services is an enduring concern of health care planners and policy-makers around the world. Rural/urban residency has long been considered as a critical determinant of health and healthcare use over time and across countries [1–3]. Over the past several decades, China has seen remarkable economic growth and improved health care. These improvements, however, are not equitable among rural and urban regions, with widely reported rural-urban differences in healthcare resources [4], health outcomes [5, 6], prevalence of diseases [7, 8], and healthcare utilization [3, 9]. For example, during the period of 1993 to 2011, urban residents in China were two to five times more likely to utilize outpatient and inpatient care than rural residents [9].

Inequality in socioeconomic status (SES) between residents in rural and urban areas of China may partially account for the rural-urban gaps in healthcare use [10]. For many decades, urban residents (defined as those living in areas under the jurisdiction of cities and towns) in China have tended to have higher household income than rural residents (defined as those living in countryside) [10], and in the past two decades urban China has seen a much faster economic growth than rural parts of the nation [11].

Health insurance may also play a significant role in healthcare use. In China, public health insurance dominates the health insurance market, and the public health insurance programs available to rural and urban residents have long been operated separately for rural and urban residents. The employment-based insurance, the Urban Employees Based Medical Insurance (UEBMI), was initiated in urban areas in 1998. The comprehensive UEBMI plan covers inpatient, outpatient, emergency room, and prescription drug expenses [12]. The Urban Residents Basic Medical Insurance (URBMI) was launched in 2007, providing coverage for urban residents without formal employment with the goal of eliminating impoverishment due to chronic or fatal diseases; the URBMI primarily covers expenses related to inpatient care [12]. The New Rural Cooperative Medical Scheme (NRCMS) was established in 2003, which provides partial coverage for all types of medical expenses, and its caps for reimbursement vary by regions and local economic development levels [12]. In 2008, the insurance rates in China were about 65% and 90% in urban and rural regions, respectively [12, 13].

In 2009, China launched an aggressive and comprehensive healthcare reform aimed to achieve affordable and equitable healthcare for all by 2020, with an estimated CNY850 billion (about US $124 billion) governmental investment [14–16]. In 2011, 97% of rural and 95% of urban residents enrolled in public health insurance programs (i.e., the UEBMI, the URBMI, or the NRCMS) [17], indicating the establishment of universal health insurance coverage. To maintain the universal coverage, China's government increased per capita subsidies for public health insurance premiums by 60% from 2011 to 2014 [18]. In 2012, China expanded health insurance coverage for critical illness (e.g., lung cancer) without increasing premiums to improve covered insurance benefits and reduce personal catastrophic healthcare spending. In 2014, 700 million people were covered by the critical illness insurance, under a total of CNY9.7 billion ($1.6 billion) funds reserved for this program [19].

China has the largest older population (age 65 or over) among the developing countries [20]; by 2027, its older population will increase to 20% (from 7% in 2002) [21]. Population ageing raises concerns about availability of healthcare services, increased healthcare costs, and sustainability of China's pension system [9]. These concerns may be more pronounced for rural older adults who tend to have less access to care and less stable income than urban older adults, despite recent improvements in health insurance coverage.

Previous studies documented significant rural-urban gaps in healthcare and health outcome measures [3, 21–33], although most studies focused on measures for all adults in China

rather than older adults, and several studies only reported crude rural-urban differences without controlling for patient characteristics, such as demographics and disease diagnoses. Other research evaluated rural-urban differences in healthcare access among older adults in China. For example, using the Chinese Longitudinal Healthy Longevity Surveys (CLHLS), one study [25] found that the associations between access to healthcare and health outcomes were generally stronger for older residents in rural areas than in urban areas, and another study [26] that explored the impact of medical insurance on rural-urban gaps in healthcare use revealed that urban older adults had significantly better access to care and had higher healthcare expenditures than their rural counterparts. Feng and colleagues utilized the China Health and Nutrition Survey data from 1991 through 2011 and found that, compared with urban older persons, rural groups had lower medical expenditures [33]. However, these studies did not examine rural-urban differences in healthcare measures comprehensively, especially among older adults with insurance. Recent studies [27–32] evaluated the rural-urban gaps in healthcare metrics under universal health coverage. Nevertheless, their findings were either based on rural and urban residents in a single area [27, 28], or on cross-sectional analyses on older adults for a single or several selected indicators [30–32]. In an analysis of seven targeted provinces in China, Weng and Ning [29] showed that inequality in reimbursement rates of the basic medical insurance played a significant role in rural-urban differences in healthcare expenses among all insured people instead of insured older adults.

To date, little is known about the rural-urban differences in health and healthcare measures after the establishment of the universal health insurance coverage in China in 2011, especially among older adults. This study reports an overall pattern of rural-urban differences in a set of health and healthcare measures in 2011 and 2014, and compares these differences between the two years to track possible changes over time.

## Materials and methods

### Data sources

This study used data from the 2011 and 2014 waves of the CLHLS. The CLHLS is the first national survey conducted in 631 randomly selected counties and cities in 22 of the 31 provinces in China, covering about 85% of the total population [34]. It provides self-reported information on activities of daily living (ADLs), instrumental ADL (IADLs), healthcare utilization, healthcare expenditures, demographics, family and household characteristics, lifestyle, psychological characteristics, and economic resources for adults aged 65 or over [35]. Previous studies reported high reliability, validity, and other aspects of data quality in the CLHLS [36]. Zeng and colleagues provided more details about the CLHLS, including sampling design, follow-up interviews, procedures, and data quality [34].

### Study sample

There were 7,327 and 7,100 observations in the 2011 and 2014 waves of CLHLS, respectively. Of the 14,427 individuals in the two years data, 7,039 were identified as rural residents, and 7,388 were urban residents. Because this study focused on older adults with public health insurance (defined as the UEBMI, the URBMI, or the NRCMS), 1,747 uninsured residents were excluded. We further excluded 3,759 individuals who lived in urban areas but were covered by the NRCMS, because they were immigrants who had rural hukou (a mandatory regulation of household registration in China) but lived in urban cities. Therefore, we excluded them from the study sample because they had different access to care than other urban residents due to their rural insurance status. We conducted sensitivity analyses in which the 3,759 individuals were included in multivariable regressions; the results were very similar to results

reported in the study (S19–S21 Tables in S1 File). Our study sample included 2,624 urban and 6,297 rural residents.

## Independent variable of interest and outcomes

The independent variable of interest in this study was the rural/urban residency status. The CLHLS provides urban/rural residency at the time of survey (rather than "hukou" status). According to the methodology proposed by the National Bureau of Statistics of China [37, 38], and based on prior studies [37, 39], rural/urban residency was defined in this study by one question in the CLHLS: "What is the current residence area of the interviewee?" We coded the answers as 1 (i.e., urban area) if the answers were city or town, and 0 for a rural area.

The outcome variables included measures for health outcomes, adequate access to care, and healthcare expenditures. Health outcome measures included those for ADL, IADL, and psychological well-being. For ADL, we extracted 5 items from the CLHLS that measured levels of independence for bathing, dressing, toilet use, transferring, and eating. The IADL measure included 8 items for communication, shopping, cooking, laundry, walking continuously for 1 kilometer, lifting a weight, continuously crouching and standing up three times, and taking public transportation to assess the elders' independent living skills. Each ADL or IADL item measures functional status on a scale from 0 to 2 (assistance needed always, assistance needed sometimes, and no assistance needed, respectively). Thus, the total score ranges from 0 to 10 for the ADL measure and from 0 to 16 for the IADL measure, with a higher score indicating more independence. The measure of psychological well-being was derived from 4 items in the CLHLS and had a score ranging from 0 to 4, with a higher score indicating better psychological state (S1 Text in S1 File).

Adequate access to healthcare services, as measured by the availability of care for those who do need care [39], was defined by a single question in the CLHLS: "Could you get adequate medical service at present when it is necessary?" with possible answers of yes (coded as 1) or no (coded as 0). Furthermore, we included a set of healthcare expenditure indicators, including total expenditure, total out-of-pocket (OOP) spending, total expenditures for inpatient and outpatient care, OOP expenditures for inpatient and outpatient care, and ratio of total OOP expenditures to total expenditures (self-payment ratio). We obtained the Consumer Price Index from the National Bureau of Statistics of China, and adjusted all 2011 expenditures to the 2014 amount [40]. More details about these outcomes are described in the appendix (S1 Text in S1 File).

## Covariates

According to previous studies [34, 39, 41] on health outcomes and healthcare utilization, we extracted relevant covariates from the CLHLS including individual demographics, SES in childhood and at present, family care resources, and health behaviors. Demographic information included age groups (65–69, 70–79, 80–89, 90–99, > = 100) and sex (male/female). Childhood SES was measured by whether the respondent went to bed hungry (yes, no, and missing) and got adequate medical services when sick (yes, no, and missing) in childhood. Current SES was measured by education level (never, elementary school, middle school, high school or higher, and missing) and occupation (profession/administration, others, and missing). Family care resources included marital status (married/single), whether the respondent was living with others (yes/no), the number of living children, whether the respondent had sufficient financial support for daily costs (yes/no), and annual income per capita. Health behavior measures included those about smoking status, alcohol drinking behavior, exercise, sleep quality, and regular physical examination. We included regional dummies (east, middle, and west) to

adjust for possible geographic variations. We included arm length as an indicator of early-life nutritional status [42], which has been considered a preferred anthropometric measure for studies of the elderly [43–45]. In multivariable analyses for healthcare expenditures (and self-payment ratio), we also adjusted for the following covariates: self-reported health (very good, good, so-so, bad), whether the respondent had serious illness in the last 2 years, the number of diagnosed chronic diseases, scores of ADL, IADL and psychological well-being, and cognitive function measured by the Mini Mental State Examination score [39, 46].

## Statistical analysis

We first compared health outcomes, healthcare use and expenditures, and covariates between rural and urban residents, pooling the 2 waves of data (2011 and 2014). We used $\chi^2$ tests for categorical variables, and t tests for continuous variables for comparisons.

We fitted multivariable regression models on the pooled data, using linear regression for continuous health outcome variables (ADL, IADL and psychological well-being scores), and a logit regression for the binary dependent variable of adequate access to care.

The health expenditures data included nonnegative values, with a substantial proportion of the values being zero. In a review study, Mihaylova and colleagues recommended that the two-part model be used for modeling expenditure data with excessive zeros [47]. The two part model, with logit or probit in the first part and a generalized linear model (GLM) in the second part, has also been widely used in recent health service research studies [48–51]. In the present study, we fitted two-part models for all expenditure variables with a logit model in the first part, modeling if the respondent had positive expenditure, and a GLM with gamma distribution and log link function in the second part, modeling patterns of positive expenditures. Because urban residence was a time invariant variable, multivariable regressions with random effects were applied to all measures.

We further fitted the same multivariable regression models above on each of the 2011 and 2014 waves of data separately. We then conducted a nonparametric test with bootstrap resampling (500 times) to compare the coefficients for rural-urban differences in 2011 and 2014.

Education, occupation, whether respondents went to bed hungry, or had sufficient medical service in childhood had relatively high missing rates, ranging from 4.4% to 20.6%. We defined missing values as a separate group in the main analyses (described above). In the sensitivity analyses, we excluded the individuals with any missing values, and the results remained very similar and thus are not reported. All regressions reported robust standard error.

To help ease the interpretation of model results, we computed margins of adjusted outcomes for urban (i.e., Urban-adjusted in Tables 2 and 3) and rural (i.e., Rural-adjusted in Tables 2 and 3) residents, respectively, by applying the "margins" STATA command after multivariable regressions; the marginal estimates of rural-urban differences in outcomes were obtained in a similar way. We used STATA version 15.1 (Stata Statistical Software: Release 15. College Station, TX: StataCorp LLC) for statistical analyses. Since our study tested 11 outcomes, the Bonferroni correction method was used to adjust for multiple comparisons [52]. Therefore, the threshold of a corrected P value for statistical significance was 0.005.

## Ethics statement

Our study was approved by the Research Subjects Review Board of the University of Rochester.

## Results

Table 1 presents the descriptive statistics of respondent characteristics by urban/rural residency. Urban residents were more dependent in ADL functions, but had better psychological well-being than rural residents. Urban residents had higher total and OOP expenditures for inpatient care, outpatient care, and all health care, but had lower self-payment ratios than rural residents. Urban residents also reported having greater adequate access to care than rural residents.

After adjusting for covariates, rural-urban differences in health measures were still significant (**Table 2 and S1–S6 Tables in** S1 **File**). Urban residents were more dependent on ADLs (adjusted difference = -0.62; P < 0.0001) and IADLs (adjusted difference = -1.24; P < 0.0001) and reported greater access to care (adjusted odds ratio = 2.24; P = 0.0018). Urban residents also had higher adjusted total expenditures for inpatient care (adjusted difference = CNY1475; P < 0.0001), outpatient care (adjusted difference = CNY1338; P < 0.0001), and both inpatient and outpatient care (adjusted difference = CNY2730; P < 0.0001), as well as higher adjusted OOP expenditures for outpatient care (adjusted difference = CNY406; P < 0.0001), and inpatient and outpatient care combined (adjusted difference = CNY857; P < 0.0001). We also found that urban residents were more likely to have lower self-payment ratios (adjusted difference = -13.7%; P < 0.0001) than their rural counterparts. There were no significant differences in psychological well-being (adjusted difference = 0.06; P = 0.0220) and total inpatient OOP spending (adjusted difference = CNY379; P = 0.0051) between rural and urban residents.

In analyses stratified by year, we found slightly improved ADL and IADL functions, psychological well-being, adequate access to care, healthcare expenditures (higher), and self-payment ratio (lower) for both rural and urban residents from 2011 to 2014 (Table 3, S7–S18 Tables, and S1 Fig in S1 File). Although urban and rural residents were not significantly different in adequate access to care, total OOP expenditures for inpatient, outpatient, and total care in 2011 or in psychological well-being and adequate access to care in 2014, urban and rural residents significantly differed in most other health measures in the two years.

Our results also demonstrated that the gaps in health outcomes, adequate access to care, and self-payment ratio between rural and urban residents narrowed, but differences in healthcare expenditures were exacerbated from 2011 to 2014. Table 4 reports the nonparametric comparisons of the adjusted rural-urban differences between 2011 and 2014. We found that rural-urban differences significantly decreased in ADLs (change in rural-urban difference = -0.07; P < 0.0001), IADLs (change in rural-urban difference = -0.18; P < 0.0001), psychological well-being (change in rural-urban difference = 0.10; P < 0.0001), adequate access to care (change in rural-urban difference = 1.11; P < 0.0001), and self-payment ratio (change in rural-urban difference = -11.7%; P < 0.0001). However, rural-urban differences significantly increased in total OOP expenditures for total (change in rural-urban difference = CNY-1116; P = 0.0007), and outpatient (change in rural-urban difference = CNY-676; P = 0.0002) care from 2011 to 2014. There was no significant change in rural-urban difference in total medical (change in rural-urban difference = CNY-1065, P = 0.1055), inpatient (change in rural-urban difference = CNY-315; P = 0.5506), outpatient expenditures (change in rural-urban difference = CNY-795; P = 0.0147), and total inpatient OOP payments (change in rural-urban difference = CNY-641; P = 0.0364).

## Discussion

In this study of older adults in China with public health insurance, we evaluated the adjusted rural-urban differences in health outcomes (i.e., ADL, IADL, and psychological well-being), adequate access to care, and healthcare expenditures in 2011 and 2014. We found that urban

**Table 1. Descriptive statistics for study variables, by urban and rural residency.**

| Outcomes | Total (n = 8921) | Urban (n = 2624) | Rural (n = 6297) | P value* |
|---|---|---|---|---|
| | Mean±SD or Number ±Prevalence (%) | | | |
| ADL[a] | 8.92(2.36) | 8.69(2.61) | 9.01(2.24) | <0.0001 |
| IADL[b] | 10.67(6.02) | 10.78(6.23) | 10.63(5.94) | 0.2621 |
| Psychological well-being | 3.52(0.81) | 3.65(0.70) | 3.46(0.84) | <0.0001 |
| Adequate access to care | 8483(95.7%) | 2565(98.4%) | 5918(94.5%) | <0.0001 |
| Total medical expenditure | 4579 (12982.61) | 8529 (18769.23) | 2891 (8974.61) | <0.0001 |
| Total inpatient expenditure | 2881(9901.21) | 5201(14031.81) | 1859 (7147.76) | <0.0001 |
| Total outpatient expenditure | 1911(6271.77) | 3627 (9355.54) | 1182 (4132.92) | <0.0001 |
| Total out-of-pocket expenditure | 2038 (5757.34) | 3332 (7913.48) | 1486 (4423.76) | <0.0001 |
| Total inpatient out-of-pocket expenditure | 1466 (5530.98) | 2184 (7252.01) | 1051 (4176.41) | <0.0001 |
| Total outpatient out-of-pocket expenditure | 1118 (3107.15) | 1646 (3689.18) | 896 (2797.70) | <0.0001 |
| Self-payment ratio | 0.66(0.36) | 0.53(0.38) | 0.72(0.34) | <0.0001 |
| **Covariates** | | | | |
| Age | | | | |
| 65–69 | 433(4.8%) | 116(4.4%) | 317(5.0%) | |
| 70–79 | 2681(30.1%) | 924(35.2%) | 1757(27.9%) | |
| 80–89 | 2678(30.0%) | 760(29.0%) | 1918(30.5%) | |
| 90–99 | 2132(23.9%) | 612(23.3%) | 1520(24.1%) | |
| > = 100 | 997(11.2%) | 212(8.1%) | 785(12.5%) | <0.0001 |
| Sex | | | | |
| Female | 4615(51.7%) | 1164(44.4%) | 3451(54.8%) | <0.0001 |
| Marital status | | | | |
| Married | 3594(40.6%) | 1250(47.8%) | 2344(37.5%) | <0.0001 |
| Number of living children | 3.76(1.72) | 3.46(1.60) | 3.88(1.75) | <0.0001 |
| Annual income per capita | 10984.27(13488.55) | 18618.78(15374.77) | 7787.901(11160.60) | <0.0001 |
| Education | | | | |
| Never | 4738(53.1%) | 815(31.1%) | 3923(62.3%) | |
| Elementary school | 2853(32.0%) | 1015(38.7%) | 1838(29.2%) | |
| Middle school | 349(3.9%) | 185(7.1%) | 164(2.6%) | |
| High school or higher | 584(6.6%) | 436(16.5%) | 148(2.4%) | |
| Missing | 397(4.4%) | 173(6.6%) | 224(3.5%) | <0.0001 |
| Living with people | | | | |
| Yes | 7370(83.1%) | 1321(88.7%) | 5049(80.7%) | <0.0001 |
| Drinking at present | | | | |
| Yes | 1456(16.5%) | 392(15.1%) | 1064(17.1%) | 0.0199 |
| Smoking at present | | | | |
| Yes | 1567(17.6%) | 428(16.4%) | 1139(18.2%) | 0.0433 |
| Regular exercise at present | | | | |
| Yes | 2979(33.9%) | 1462(56.5%) | 1517(24.4%) | <0.0001 |
| Sufficient financial support | | | | |
| Yes | 7249(81.7%) | 2342(89.5%) | 4907(78.4%) | <0.0001 |
| Went to bed hungry in childhood | | | | |
| No | 2118(23.7%) | 931(35.5%) | 1187(18.8%) | |
| Yes | 5992(67.2%) | 1581(60.3%) | 4411(70.1%) | |
| Missing | 811(9.1%) | 112(4.2%) | 699(11.1%) | <0.0001 |
| Able to access to healthcare in childhood | | | | |
| No | 4456(50.0%) | 1191(45.4%) | 3265(51.9%) | |

*(Continued)*

**Table 1.** (Continued)

| Outcomes | Total (n = 8921) | Urban (n = 2624) | Rural (n = 6297) | P value* |
|---|---|---|---|---|
| | Mean±SD or Number ±Prevalence (%) | | | |
| Yes | 2627(29.4%) | 1139(43.4%) | 1488(23.6%) | |
| Missing | 1838(20.6%) | 294(11.20) | 1544(24.5%) | <0.0001 |
| Quality of sleeping | | | | |
| Very good | 1661(18.7%) | 655(25.0%) | 1006(16.0%) | |
| Good | 3843(43.2%) | 991(37.8%) | 2852(45.4%) | |
| So-so | 2299(25.8%) | 633(24.2%) | 1666(26.5%) | |
| Bad | 1101(12.3%) | 339(13.0%) | 762(12.1%) | <0.0001 |
| Arm length | 50.77(7.93) | 51.47(8.96) | 50.48(7.44) | <0.0001 |
| Number of diagnosed chronic diseases | 2.49(4.83) | 3.39(5.33) | 2.11(4.56) | <0.0001 |
| Severe disease | | | | |
| Yes | 2240(25.8%) | 917(35.7%) | 1323(21.4%) | <0.0001 |
| Occupation | | | | |
| Profession/ Administration | 882(9.9%) | 694(26.5%) | 188(3.0%) | |
| Others | 7521(84.3%) | 1909(72.7%) | 5612(89.1%) | |
| Missing | 518(5.8%) | 21(0.8%) | 497(7.9%) | <0.0001 |
| Regular physical examination | | | | |
| Yes | 4163(47.0%) | 1123(42.9%) | 3040(48.7%) | <0.0001 |
| MMSE[c] | 22.85(8.86) | 24.19(8.42) | 22.29(8.98) | <0.0001 |
| Self-reported health | | | | |
| Very good | 823(9.3%) | 338(12.9%) | 485(7.7%) | |
| Good | 2984(33.5%) | 873(33.4%) | 2111(33.6%) | |
| So-so | 3193(35.9%) | 912(34.8%) | 2281(36.3%) | |
| Bad | 1900(21.3%) | 496(18.9%) | 1404(22.4%) | <0.0001 |
| Region | | | | |
| East | 4268(47.8%) | 1288(49.1%) | 2980(47.3%) | |
| Middle | 2594(29.1%) | 628(23.9%) | 1966(31.2%) | |
| West | 2059(23.1%) | 708(27.0%) | 1351(21.5%) | <0.0001 |

SD = standard deviation.

[a]Activity of daily living.

[b]Instrumental activity of daily living.

[c]Mini-mental State Examination.

*$\chi^2$ tests for categorical variables, and t tests for continuous variables between rural and urban.

residents had worse physical health status, more access to care, higher healthcare expenditures, and lower self-payment ratios compared to rural residents. Rural-urban differences in health outcomes, adequate access to care, and self-payment ratio significantly decreased, while the differences in healthcare expenditures significantly increased from 2011 to 2014.

Our findings that urban residents had worse physical function than their rural counterparts are consistent with results of previous studies [10, 31, 53–57]. Several potential explanations are provided for our results. First, recent economic development in China might have exposed urban residents to higher air and water pollution than rural residents [58], limiting urban residents' outdoor activities and reducing their physical functional ability. Furthermore, recent studies [54, 59] have demonstrated that the decreased physical functional ability among older urban residents was significantly associated with air pollution. Second, population density in urban China is extremely high, and as a result a large majority of urban residents live in

**Table 2. Multivariable regression analyses based on pooled 2011 and 2014 data.**

| Outcomes | Urban-adjusted[c] | Rural-adjusted[d] | Adjusted difference[e] | P value |
|---|---|---|---|---|
| ADL[a] | 8.52 | 9.14 | -0.62 | <0.0001 |
| IADL[b] | 9.84 | 11.08 | -1.24 | <0.0001 |
| Psychological well-being | 3.57 | 3.51 | 0.06 | 0.0220 |
| Adequate access to care[f] | 0.99 | 0.98 | 2.24 | 0.0018 |
| Total medical expenditure | 6335 | 3605 | 2730 | <0.0001 |
| Total inpatient expenditure | 3793 | 2318 | 1475 | <0.0001 |
| Total outpatient expenditure | 2708 | 1370 | 1338 | <0.0001 |
| Total out-of-pocket t expenditure | 2575 | 1718 | 857 | <0.0001 |
| Total inpatient out-of-pocket expenditure | 1648 | 1269 | 379 | 0.0051 |
| Total outpatient out-of-pocket expenditure | 1381.34 | 975.54 | 405.81 | <0.0001 |
| Self-payment ratio | 55.8% | 69.5% | -13.7% | <0.0001 |

[a]Activity of daily living.

[b]Instrumental activity of daily living.

[c,d]Urban-adjusted and rural-adjusted columns report margins of adjusted outcomes.

[e]Adjusted differences are marginal differences calculated based on the coefficients of the Urban variable.

[f]The adjusted difference of adequate access to care is odds ratio.

apartment buildings. The elderly who live in apartments either take elevators or live on the ground floor, and very few have access to yards or gardens [53]. Therefore, the amount of physical activities that the older Chinese urban population participated in might be reduced, subsequently resulting in increases in physical limitations [10, 53]. While the majority of the older Chinese rural adults live in houses, and they have their own garden, and/or agricultural

**Table 3. Multivariable regressions by year.**

| Outcomes | 2011 | | | | 2014 | | | |
|---|---|---|---|---|---|---|---|---|
| | Urban-adjusted[c] | Rural-adjusted[d] | Adjusted difference[e] | P value | Urban-adjusted[f] | Rural-Adjusted[g] | Adjusted difference[h] | P value |
| ADL[a] | 8.47 | 9.13 | -0.66 | <0.0001 | 8.59 | 9.18 | -0.59 | <0.0001 |
| IADL[b] | 9.73 | 11.11 | -1.38 | <0.0001 | 9.98 | 11.18 | -1.20 | <0.0001 |
| Psychological well-being | 3.57 | 3.47 | 0.10 | 0.0029 | 3.54 | 3.54 | -0.00 | 0.9360 |
| Adequate access to care[i] | 0.99 | 0.97 | 2.13 | 0.0080 | 0.99 | 0.98 | 1.93 | 0.0848 |
| Total medical expenditure | 5536 | 3192 | 2344 | <0.0001 | 7343 | 3934 | 3409 | <0.0001 |
| Total inpatient expenditure | 3255 | 1967 | 1288 | <0.0001 | 4284 | 2681 | 1603 | <0.0001 |
| Total outpatient expenditure | 2365 | 1336 | 1029 | <0.0001 | 3200 | 1376 | 1824 | <0.0001 |
| Total out-of-pocket t expenditure | 2247 | 1887 | 360 | 0.0193 | 3050 | 1574 | 1476 | <0.0001 |
| Total inpatient out-of-pocket expenditure | 1317 | 1156 | 161 | 0.2770 | 2246 | 1444 | 802 | 0.0008 |
| Total outpatient out-of-pocket expenditure | 1215 | 1092 | 123 | 0.2062 | 1660 | 861 | 799 | <0.0001 |
| Self-payment ratio | 56.6% | 76.2% | -19.6% | <0.0001 | 55.5% | 63.4% | -7.9% | <0.0001 |

[a]Activity of daily living.

[b]Instrumental activity of daily living.

[c,d,f,g]Urban-adjusted and rural-adjusted columns report margins of adjusted outcomes.

[e,h]Adjusted differences are marginal differences calculated based on the coefficients of the Urban variable.

[i]The adjusted difference of adequate access to care is odds ratio.

**Table 4. Nonparametric test results.**

| Outcomes | Change in rural-urban difference[c] (2011 vs 2014) | P value (Nonparametric tests) |
|---|---|---|
| ADL[a] | -0.07 | <0.0001 |
| IADL[b] | -0.18 | <0.0001 |
| Psychological well-being | 0.10 | <0.0001 |
| Adequate access to care[d] | 1.11 | <0.0001 |
| Total medical expenditure | -1065 | 0.1055 |
| Total inpatient expenditure | -315 | 0.5506 |
| Total outpatient expenditure | -795 | 0.0147 |
| Total out-of-pocket expenditure | -1116 | 0.0007 |
| Total inpatient out-of-pocket expenditure | -641 | 0.0364 |
| Total outpatient out-of-pocket expenditure | -676 | 0.0002 |
| Self-payment ratio | -11.7% | <0.0001 |

[a]Activity of daily living.

[b]Instrumental activity of daily living.

[c]Change in rural-urban difference = Adjusted difference in 2011 −Adjusted difference in 2014.

[d]Change in rural-urban difference of adequate access to care is odds ratio (Change in rural-urban difference in coefficient of access to care = 0.102).

field [53]. They perform garden work to grow vegetables or even perform regular labor in the fields, which contributes to maintained capacity for daily living activities [53]. In addition, it is very common that older Chinese rural persons continue to work until ages 60–69 years, and the percentage of older working adults in ruran areas declines to below 20% only after 80 years old [60, 61]. Third, in general, Chinese rural residents may value independence more highly than urban residents [10, 53, 62]; thus, older rural residents may be more proactive to maintain physical activities and their physical and functional independence.

Several other studies, however, reported somewhat different results about the rural-urban difference in physical function. Using two waves of data from the China Sampling Surveys on Disability, Peng and colleagues concluded that urban residents had better physical abilities than rural residents in the analysis of sampled persons aged 0 to 85 (or above) [24]. Two other studies [63, 64] using the China Health and Retirement Longitudinal Study database reported that urban residents had lower risk of physical disability than rural groups among people aged between 45 and 80. These different findings may be due to the different samples included in these studies (e.g., the trajectories of physical function and disability may be different among adolescents, middle-aged adults, and older adults), different analytic approaches (e.g., one study [24] did not adjust for patient characteristics as possible confounders, and another study [64] used projected estimates to compare future rural-urban difference), and different research questions being tested (e.g., Hou and colleagues [63] examined the effects of urbanization on health status by comparing health measures among residents in recently urbanized areas, rural areas, and existing urban areas).

Recent economic development in China may have benefited residents in both urban and rural areas, which could explain the improved physical function from 2011 to 2014 among both groups. The annual average per capita disposable income rose from CNY6977 [65] in 2011 to CNY10489 [66] in 2014 in rural China, and from CNY21810 [65] in 2011 to CNY29381 [66] in 2014 in urban areas; increased disposable income, especially among urban residents, may make paid outdoor activities more affordable. China's economic development also enables urban and rural communities to provide more facilities for older residents

(especially for older urban residents with limited physical activities). Moreover, both rural and urban residential committees organized diverse activities (e.g., group dancing), encouraging the elderly to be more physically active.

Differences in SES were reported to be a significant factor explaining different psychological health status among Chinese older people [67]. The findings of improved psychological well-being among rural residents and narrowed rural-urban differences from 2011 to 2014 may be explained by the faster increase rate in annual average per capita disposable income among rural residents [65, 66, 68]. The improved psychological health status among rural residents may also result from the continuous expansion and improved benefits of public health insurance in rural areas. Publicly financed insurance covers outpatient and inpatient mental health care [69], including diagnosis, treatment, and rehabilitation services [69], and as a result, rural residents had more access to mental health care over time.

In line with earlier studies [7, 9, 70], our study showed that urban residents had significantly better access to care than rural residents. People residing in rural areas usually suffer from a shortage of healthcare providers, extended travel to health care facilities, lower income to purchase health services, and lack of social support [71, 72]. Financing for China's health care institutions partially depends on local governments, which vary considerably in their financial capacities between well-developed urban areas and under-developed rural villages. The number of village health clinics increased by only 8% from 2005 to 2017, whereas the number of hospitals in urban areas grew by 66% over the same period [73]. It has been reported that urban–rural disparities in supply of healthcare providers account for about a third of overall inter-county inequality [74]. Different health insurance benefits may be another reason for self-reported disparities in access to care [75]. Rural residents are stipulated to participate in the local NRCMS, which has less comprehensive benefits than that of the UEBMI and URBMI programs available for urban residents. About 53.4% of hospitalization expenditures of older people in urban areas and 30.5% in rural areas were reimbursed by medical insurance in 2012 [67]. Under the two-tiered health insurance systems, rural residents usually encounter more financial barriers to healthcare, although our results suggest that rural-urban disparities in self-reported access to care narrowed slightly from 2011 to 2014. The narrowed disparities over time likely reflect the faster economic growth rate in rural areas and targeted efforts of China's government to improve insurance coverages for rural residents in recent years. In line with these findings on self-reported access to care and potential explanations, we further found that, although urban residents had significantly lower self-payment ratios than rural residents over time for healthcare, this rural-urban difference was reduced substantially from 2011 to 2014.

Similar to previous research [76], our study revealed increasing gaps in healthcare expenditures for both inpatient and outpatient care between urban and rural residents, despite the reduced rural-urban disparities in self-reported access to care. This suggests that although rural residents experienced significant improvements in insurance coverage and perceived access to care, urban residents benefited disproportionately from increased insurance subsidies, improved insurance coverages, and overall economic growth in terms of realized access to health care after adjusting for differences in physical and mental health conditions, as well as diagnoses of chronic conditions.

Our study had several limitations. First, this study was not able to examine the causes of the rural-urban differences. Although we discussed several possible explanations above, it is possible that other factors, such as physician/provider practice styles and environmental factors, are also related to health status, healthcare utilization, and healthcare expenditures, as well as rural-urban differences in these measures. Examining how these factors may be related to rural-urban differences will be important research areas for further study. Second, our study

relies on self-reported measures of health outcome, healthcare utilization, and expenditure, which leads to potential recall bias in survey responses, and which may bias the estimated rural-urban differences if urban and rural residents differed in how they responded to survey questions. Third, we were not able to control for individual fixed effects in the pooled analysis because different persons were sampled in the surveys of 2011 and 2014; and we were also unable to determine persons who might appear in both years' surveys because the data we had do not allow us to identify them. Fourth, we did not specifically conduct analyses on China's rural-urban differences associated with different types of medical insurance initiatives. The UEBMI, URBMI, and NRCMS are significantly different from each other in terms of covered benefits and beneficiary characteristics, and future studies should be conducted to compare the three health insurance schemes separately, and evaluate the extent to which they contribute to China's rural-urban differences in health outcomes and expenditures.

## Conclusions

In conclusion, this study found that health outcomes and self-reported access to care improved from 2011 to 2014 for both rural and urban older adults in China, and rural-urban differences narrowed. However, rural-urban differences in health care expenditures increased from 2011 to 2014, despite growing expenditures in both groups. The remaining urban-rural differences are possibly due to variations in health insurance coverages, available healthcare resources, and economic development between rural and urban areas. Our findings provide evidence that supports China's implementation of integrated rural and urban public health insurance systems staring in 2019. Additionally, inequalities in the healthcare resource distribution and economic development between rural and urban areas should be addressed.

## Supporting information

**S1 File.**
(DOCX)

**S1 Data.**
(ZIP)

## Acknowledgments

This research uses data from Chinese Longitudinal Healthy Longevity Surveys (CLHLS). We thank the China Center for Economic Research at Peking University for supplying the CLHLS data from 2011 to 2014.

## Author Contributions

**Conceptualization:** Meiling Ying, Yue Li.

**Data curation:** Meiling Ying, Sijiu Wang, Chen Bai.

**Formal analysis:** Meiling Ying, Chen Bai.

**Methodology:** Meiling Ying, Sijiu Wang, Yue Li.

**Supervision:** Yue Li.

**Writing – original draft:** Meiling Ying.

**Writing – review & editing:** Meiling Ying, Sijiu Wang, Chen Bai, Yue Li.

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
