## [Decision Letter · Decision Letter 0]

23 Jul 2020

PONE-D-20-10022

Rural-Urban Differences in Health Outcomes, Healthcare Use, and Expenditures among Older Adults under Universal Health Insurance in China

PLOS ONE

Dear Dr. ying,

Thank you for submitting your manuscript to PLOS ONE. After careful consideration, we feel that it has merit but does not fully meet PLOS ONE’s publication criteria as it currently stands. Therefore, we invite you to submit a revised version of the manuscript that addresses the points raised during the review process.

Please respond to both reviewers' comments below and the comments of reviewer #1 attached. Thank you.

We look forward to receiving your revised manuscript.

Kind regards,

Oliver Gruebner

Academic Editor

PLOS ONE

Journal Requirements:

2. Please include your tables as part of your main manuscript and remove the individual files.

Please note that supplementary tables should be uploaded as separate "supporting information" files.

Reviewers' comments:

Reviewer's Responses to Questions

**Comments to the Author**

1. Is the manuscript technically sound, and do the data support the conclusions?

Reviewer #1: Yes

Reviewer #2: Yes

2. Has the statistical analysis been performed appropriately and rigorously? 

Reviewer #1: I Don't Know

Reviewer #2: Yes

3. Have the authors made all data underlying the findings in their manuscript fully available?

Reviewer #1: Yes

Reviewer #2: Yes

4. Is the manuscript presented in an intelligible fashion and written in standard English?

Reviewer #1: Yes

Reviewer #2: No

5. Review Comments to the Author

Reviewer #1: I found the manuscript particularly interesting and well written so I just have a few minor comments and questions (on the manuscript).

I suggest the authors to improve the abstract and to include one part that is missing on the manuscript: these results will be important for what and for who? Which proposals you have to narrow even further the differences between urban and rural?

Reviewer #2: The major issue of this manuscript is the language, and a considerable amount of careless mistakes in terms of wording, grammars, etc could be identified. For example, in abstract line 52, IADL has not been explained; in line 94, "have" should be used instead of "has"; in line 129-130, "another" instead of "the other"; in line 233, should be "fitted" instead of "fit"; in line 331, typo of "Chinses"; in line 347, should be "examine" not "examined"; a mistake appears in table A5, row 1; etc. I would strongly suggest the authors to revise the manuscript carefully to check the language, and preferably have it proofread by English native speakers since some sentences and expressions are not accurate and unclear.

The study is sound with appropriate statistical analyses, and I have two minor questions in terms of technical issues: 1. Whether and how did you deal with the multiple testing problem? 2. Have you checked the collinearity of covariates in regression models?

6. PLOS authors have the option to publish the peer review history of their article (what does this mean?). If published, this will include your full peer review and any attached files.

Reviewer #1: **Yes: **Claudia Costa

Reviewer #2: No

---

## [Author Response · Author response to Decision Letter 0]

27 Aug 2020

August 24, 2020

Dear Dr. Gruebner:

Thank you for considering our manuscript entitled “Rural-urban differences in health outcomes, healthcare use, and expenditures among older adults under universal health insurance in China” (PONE-D-20-10022) for publication in PLOS ONE. We wish to thank you and the reviewer for the comments, and the time you have spent reviewing our manuscript. The revisions we made in response to reviewers’ comments are highlighted in red font in the manuscript. Responses to these comments are also summarized below:

Editor:

We formatted our manuscript based on the PLOS ONE’s style requirements.

2. Please include your tables as part of your main manuscript and remove the individual files. Please note that supplementary tables should be uploaded as separate "supporting information" files.

Tables have been included in the manuscript. In addition, we uploaded the “supporting information” file that involved all supplementary tables and figures to the journal. 

3. Please include captions for your Supporting Information files at the end of your manuscript,

and update any in-text citations to match accordingly. Please see our Supporting

Information guidelines for more information: http://journals.plos.org/plosone/s/supporting-information

We have included the captions on pages 29 and 30 of the revised paper. 

Reviewer #1:

I found the manuscript particularly interesting and well written so I just have a few minor comments and questions (on the manuscript).

I suggest the authors to improve the abstract and to include one part that is missing on the manuscript: these results will be important for what and for who? Which proposals you have to narrow even further the differences between urban and rural?

We agree with this comment. We have edited the abstract based on the reviewer’s comments.

1. the abstract should start with the overall purpose of the study and the research problem the authors investigated

We have added the research goal/problem at the beginning of the abstract. 

2. using CNY and the number of respondents is a bit confusing, specially because the first is not always regarding the urban residents. I would suggest to find another acronym that makes esear to read, e,g, urbanCNY vs ruralCNY. Moreover, instead of using the numbers, I would suggest to use rates. Otherwise the reader must be always checking the total number of respondents to evaluate the values. including the values on the abstract makes the reading more difficult and removes the space the authors need to write what is missing: introduction and for what these results will be helpful

We have taken the suggestion of the reviewer and removed all numbers/values from the abstract and included introduction and relevant policy implications in the revised abstract. 

3. I checked this reference and in the document the author included a reference where he found this information (National Health Services Survey, 2008). It is important to include the main source of this information

We have cited the 2008 National Health Services Survey (NHSS) report (see reference #13). However, the report did not directly document the rural and urban insured rates. Therefore, we cited another study (see reference #12) that used the data from the 2008 NHSS report to calculate the insured rates for rural and urban areas, respectively. 

4. why the rates are so low among the urban residents? they are not allowed to have or they prefer other kinds of insurance?

Based on the cited research (see reference #12), the 65% insured rates in 2008 of the urban residents were calculated based on the urban public health insurance program rate, including the Urban Employee Based Medical Insurance (UEBMI) enrollment rate and the Urban Residents Basic Medical Insurance (URBMI) coverage rate. The UEBMI, which is an employment-based insurance, started in 1998; and the URBMI was a voluntary insurance program initiated in 2007.The New Rural Cooperative Medical Scheme was established in 2003. Its coverage rate expanded from 3% in 2003 to 90% in 2008. The cited paper indicated that the shorter launch period of the URBMI was the primary cause of the lower coverage rates for the urban residents. In addition to the public health insurance programs, there are other insurance programs available for urban residents in China, such as commercial health insurance. According to the NHSS report, 6.9% of urban residents were covered by the private insurance in 2008

5. I would suggest to include the share of increase, rather than the values himself

Thank you for this comment. On page 3 of the revised manuscript, we have edited the corresponding sentence. 

6. ageing, not aging

We have corrected the typo.

7. design, not deign

We have corrected the typo.

8. the uninsured resident where mainly urban or rural?

Our data showed that, among the uninsured residents, 42.5% were rural residents and the remaining 57.5% were urban residents.

9. better to explain here what hukou means

One page 6 of the revised paper, we added an explanation to define hukou in the study sample section of the revised manuscript. 

10. out-of-pocket

We have corrected the typo throughout the revised manuscript. 

11. since there are rural elders still working, maybe it would be relevant to adjust by this (working/not working)

We agree that working status might be an important risk factor. We defined a working status variable based on the question in the Chinese Longitudinal Healthy Longevity Surveys “are you still engaged in paid jobs after retirement”. However, of the 8,921 residents (i.e., the sample of our study), 355 residents are still working, and 8,566 residents did not respond to this question. Furthermore, there are only 86 rural residents working in paid jobs. Therefore, it is not possible for us to adjust for this variable in our analyses due to data limitation (for example, the large number of missing values), and we had to exclude it from the regression analyses. However, we expect that the relatively small number of rural elders still working would not substantially bias the estimated rural-urban differences in this study. 

12. since these values are on table 1, the authors don't need to write them here

We agree. We have removed the values on page 10 of the revised paper.

13. is there information about the share of urban residents that din't born in urban areas or that move to cities or towns when start to work?

The questionnaire did not provide this information unfortunately. 

14. Chineses

We have corrected the typo in the discussion section.

15. it is lacking information about this reference. I checked and it is a dissertation

We have replaced the dissertation with a peer-reviewed article as the reference (see reference #12).

16. it would be good to have a translation of these titles to be included here

We have translated the Chinese references to English. 

Reviewer #2:

 The major issue of this manuscript is the language, and a considerable amount of careless mistakes in terms of wording, grammars, etc could be identified. For example, in abstract line 52, IADL has not been explained; in line 94, "have" should be used instead of "has"; in line 129-130, "another" instead of "the other"; in line 233, should be "fitted" instead of "fit"; in line 331, typo of "Chinses"; in line 347, should be "examine" not "examined"; a mistake appears in table A5, row 1; etc. I would strongly suggest the authors to revise the manuscript carefully to check the language, and preferably have it proofread by English native speakers since some sentences and expressions are not accurate and unclear.

We have corrected all of these typos and improved the language throughout the manuscript. 

The study is sound with appropriate statistical analyses, and I have two minor questions in terms of technical issues: 1. Whether and how did you deal with the multiple testing problem? 2. Have you checked the collinearity of covariates in regression models?

Thank you for this comment. We conducted the Bonferroni test to correct for multiple comparisons. Specifically, given the 11 outcomes examined in our study, the P value threshold is 0.005 (i.e., 0.05/11). One page 10 of the revised paper, we have added the relevant sentences. In addition, we have revised our manuscript according to the corrected P value. 

We have checked the collinearity for each model. Overall, variance inflation factor values ranged from 1.02 to 2.69, confirming the validity of our model specification and that collinearity is not a concern in our regression models. 

Thank you again for the comments and suggestions and for the opportunity to re-submit our manuscript.

---

## [Decision Letter · Decision Letter 1]

11 Sep 2020

PONE-D-20-10022R1

Rural-Urban Differences in Health Outcomes, Healthcare Use, and Expenditures among Older Adults under Universal Health Insurance in China

PLOS ONE

Dear Dr. ying,

Thank you for submitting your manuscript to PLOS ONE. After careful consideration, we feel that it has merit but does not fully meet PLOS ONE’s publication criteria as it currently stands. Therefore, we invite you to submit a revised version of the manuscript that addresses the points raised during the review process.

Thank you for the revised version. Please check the comment of reviewer #2 which requires a very small change. Thank you for your work on it!

We look forward to receiving your revised manuscript.

Kind regards,

Oliver Gruebner

Academic Editor

PLOS ONE

Reviewers' comments:

Reviewer's Responses to Questions

**Comments to the Author**

1. If the authors have adequately addressed your comments raised in a previous round of review and you feel that this manuscript is now acceptable for publication, you may indicate that here to bypass the “Comments to the Author” section, enter your conflict of interest statement in the “Confidential to Editor” section, and submit your "Accept" recommendation.

Reviewer #1: All comments have been addressed

Reviewer #2: (No Response)

2. Is the manuscript technically sound, and do the data support the conclusions?

Reviewer #1: (No Response)

Reviewer #2: Yes

3. Has the statistical analysis been performed appropriately and rigorously? 

Reviewer #1: (No Response)

Reviewer #2: Yes

4. Have the authors made all data underlying the findings in their manuscript fully available?

Reviewer #1: (No Response)

Reviewer #2: Yes

5. Is the manuscript presented in an intelligible fashion and written in standard English?

Reviewer #1: (No Response)

Reviewer #2: Yes

6. Review Comments to the Author

Reviewer #1: (No Response)

Reviewer #2: Thank you for the well-revised manuscript. There is only one minor issue left in the appendix Table S5, the two column names in the first row are the same "Total outpatient out-of-pocket expenditure". I believe one of them should be "Total inpatient out-of-pocket expenditure"?

7. PLOS authors have the option to publish the peer review history of their article (what does this mean?). If published, this will include your full peer review and any attached files.

Reviewer #1: **Yes: **Claudia Costa

Reviewer #2: No

---

## [Author Response · Author response to Decision Letter 1]

18 Sep 2020

September 11, 2020

Dear Dr. Gruebner:

Thank you for reconsidering our manuscript entitled “Rural-urban differences in health outcomes, healthcare use, and expenditures among older adults under universal health insurance in China” (PONE-D-20-10022R1) for publication in PLOS ONE. We wish to thank you and the reviewers for the comments, and the time you have spent reviewing our manuscript. The revisions we made in response to reviewers’ comments are highlighted in red font in the manuscript. Responses to these comments are also summarized below:

Editor:

Thank you for submitting your manuscript to PLOS ONE. After careful consideration, we feel that it has merit but does not fully meet PLOS ONE’s publication criteria as it currently stands. Therefore, we invite you to submit a revised version of the manuscript that addresses the points raised during the review process. Thank you for the revised version. Please check the comment of reviewer #2 which requires a very small change. Thank you for your work on it!

Thank you for this comment. Please see our response to the reviewer 2 below.

Reviewer #1:

All comments have been addressed. 

We thank reviewer 1 for confirming that all prior concerns were adequately addressed.

Reviewer #2:

Thank you for the well-revised manuscript. There is only one minor issue left in the appendix Table S5, the two column names in the first row are the same "Total outpatient out-of-pocket expenditure". I believe one of them should be "Total inpatient out-of-pocket expenditure"?

We appreciate the reviewer’s recognition of our work. We have corrected the typo, and the first column name of the appendix Table S5 is “Total inpatient out-of-pocket expenditure”. 

Thank you again for the comments and suggestions and for the opportunity to re-submit our manuscript.

---

## [Editor Report · Decision Letter 2]

22 Sep 2020

Rural-Urban Differences in Health Outcomes, Healthcare Use, and Expenditures among Older Adults under Universal Health Insurance in China

PONE-D-20-10022R2

Dear Dr. ying,

We’re pleased to inform you that your manuscript has been judged scientifically suitable for publication and will be formally accepted for publication once it meets all outstanding technical requirements.

Kind regards,

Oliver Gruebner

Academic Editor

PLOS ONE
---

## [Editor Report · Acceptance letter]

2 Oct 2020

PONE-D-20-10022R2 

Rural-Urban Differences in Health Outcomes, Healthcare Use, and Expenditures among Older Adults under Universal Health Insurance in China 

Dear Dr. Ying:

I'm pleased to inform you that your manuscript has been deemed suitable for publication in PLOS ONE. Congratulations! Your manuscript is now with our production department. 

Kind regards, 

on behalf of

Dr. Oliver Gruebner 

Academic Editor

PLOS ONE